# Non-prescription acquisition of antibiotics: Prevalence, motives, pathways and explanatory factors in the Swedish population

Christian Munthe[1☯]*, Erik Malmqvist[1☯], Björn Rönnerstrand[2☯]

**1** Department of Philosophy, Linguistics and Theory of Science, and the Centre for Antibiotic Resistance Research (CARe), University of Gothenburg, Gothenburg, Sweden, **2** The SOM Institute, and the Centre for Antibiotic Resistance Research (CARe), University of Gothenburg, Gothenburg, Sweden

☯ These authors contributed equally to this work.
* christian.munthe@gu.se

**Data Availability Statement:** The data underlying this study has been collected under the legally mandatory protection and restrictions of the Swedish Ethical Review Act (SFS-2003:460) and

## Abstract

Prior studies indicate prevalence of unregulated non-prescription use of antibiotics also in the northern European countries. The aim of this study is to investigate the extent to which antibiotics are acquired without prescription in Sweden, and people's attitudes and motives linked to this practice. We use data from an online survey of a representative sample of the Swedish general population which included questions about respondents' antibiotic use, attitudes towards antibiotics, health care contacts, self-rated health and trust in health care. We also asked about their reason for obtaining/not obtaining antibiotics without a prescription. The results show that, in the last five years, 2,3% of the respondents had acquired antibiotics in other ways than from a Swedish physician having issued a prescription, and 4,3% reported that they are likely to do so in the future. We also show that the two most important reasons for non-prescription acquisition were physicians' refusal to prescribe antibiotics followed by travel abroad. The most important reason for not obtaining antibiotics without a prescription was to not contribute to antibiotic resistance. Using logistic regression, we show that non-prescription acquisition of antibiotics, and the intention to engage in this practice in the future, are strongly associated with low trust in health care.

## 1. Introduction

How people acquire pharmaceuticals is of a general interest from a drug safety perspective [1–3]. However, the specific case of antibiotics places this question in the acutely pressing context of antimicrobial (especially antibiotic) resistance (AMR). AMR is considered a major threat to global health and the progress of modern medicine, increasingly causing significant harm in terms of human mortality and morbidity, and global economic and social development [4, 5]. In the global action plan on AMR, launched by the World Health Organization in 2015, antibiotic stewardship (AS) is a main strategic avenue, besides AMR surveillance and boosting of antimicrobial innovation [6]. AS signifies a battery of measures aimed to manage AMR. A core component is to rationalize the use of antimicrobial drugs, with tighter control over

the European Union General Data Protection Regulation (GDPR). Since the data contains sensitive personal information about persons in the European Union (as defined by GDPR), access to this data legally requires authorization by the Swedish Ethical Review Authority ("Etikprövningsmyndigheten"), EPM. To apply for such authorization, please contact EPM by post at: Etikprövningsmyndigheten, Box 2110, 750 02 Uppsala, SWEDEN, or by email at: registrator@etikprovning.se. More information about the process to receive authorization to access legally protected research data in Sweden can be found at the EPM website: https://etikprovningsmyndigheten.se/.

**Funding:** Christian Munthe, no award no., UGOT Challenges Initiative, URL: https://www.gu.se/en/research/ugot-challenges The funders had no role in study design, data collection and analysis, decision to publish, or preparation of the manuscript.

**Competing interests:** The authors have declared that no competing interests exist.

prescription as the main tool for outpatient care and use among the general public [6]. There are well-known structural problems in effecting and implementing such prescription regimes even when a mandatory prescription system is in place. For instance, a study in Catalonia in 2008 indicated a very high willingness of pharmacists to supply antibiotics without prescription, despite a legally strict prescription mandate [7] and ensuring that doctors adhere to prescription standards when patients request antibiotics is known to be challenging [8–10], linking to a general connection between antibiotic consumption volumes and corruption levels [11].

However, even if no such implementation problems would be present, the strategy assumes that reduction of antibiotic *prescription* will significantly reduce *access* to and *use* of antibiotics. This link is usually assumed in research on AMR and AS, e.g., in large studies of antibiotics use [12, 13] and major studies supporting the design of AS regimens [14]. However, various avenues for the general public to acquire antibiotics without prescription exist, e.g., sharing between friends or relatives, private import from countries with lax or no prescription mandates for antibiotics, or purchase from offshore online pharmacies evading regulatory prescription mandates. Such behaviors potentially limit the effectiveness of prescription control as a tool for AS. They also complicate the assessment of this tool because surveying prescription data may underestimate antibiotics use and overestimate the effectiveness of AS prescription control. It is therefore a crucial task for research related to AS and the general assessment and design of AMR policy to clarify the extent of non-prescription acquisition of antibiotics.

Previous research has examined self-medication with (i.e., non-prescription use of) antibiotics, indicating that the prevalence of this phenomenon varies greatly across countries. A 2006 survey of populations in 19 European countries found that between 1 and 210 in 1000 had self-medicated using antibiotics in the last 12 months, with higher rates in eastern and southern Europe than in northern and western Europe [15, 16]. Similarly, in a 2018 Eurobarometer survey the proportion of respondents stating that their last antibiotics course was not obtained from a health care professional ranged from 1% (the Netherlands) to 15% (Romania and Austria) [17]. A recent review of the US literature found that the prevalence of self-medication ranged from 1% to as high as 66% in certain populations [18], and on a global level the variation is even more striking [19]. The intention to self-medicate in the future is subject to similar variation [15, 18].

Studies on self-medication (or non-prescription use) typically define this phenomenon as the use of antibiotics obtained without prescription from a health care professional, and thus indirectly shed light on non-prescription acquisition. However, non-prescription *use* and non-prescription *acquisition* of antibiotics must be distinguished. A drug may be obtained without prescription by one person (e.g., a parent) and then used by another (e.g., a child), a situation not captured in surveys on non-prescription use, potentially leading to underestimation of this phenomenon. Moreover, non-prescription antibiotic acquisition raises concern even if the drug is not imminently used, because it may be saved for future use or inappropriately disposed of, potentially contributing to resistance-driving pollution [20]. Therefore, non-prescription acquisition of antibiotics needs attention in its own right.

Few studies of non-prescription acquisition of prescription drugs focus specifically on antibiotics [1]. A recent study of a representative sample of the Norwegian population found that 1.5% had purchased antibiotics without a prescription during travel abroad. Non-prescription purchase was associated with younger age, female gender, number of travels, occurrences of diarrhoea, and domestic antibiotic use [21]. By contrast, a Swedish pilot study of a non-representative sample of 500 people found that 20% had acquired antibiotics in other ways than from a Swedish physician issuing a prescription. Motives and pathways varied, but foreign travel or visits and online pharmacies were commonly stated sources, while dissatisfaction

with or distrust in the health care system was the most common motive disclosed [22]. These findings, together with the dearth of research in this area and documented pathways for non-prescription access to medicines in general [3, 23], give cause for concern and provide strong reason to study the matter more closely.

A limitation of previous studies on non-prescription antibiotic acquisition concerns the explanatory factors used in the analysis–mainly socioeconomic/demographic factors and health-related behaviors [21]. This limited focus makes it difficult to draw broader conclusions regarding different societies' ability to counter non-prescription antibiotics acquisition or connect findings to a general discussion about the prospect for legitimacy and compliance in relation to public health policy regulations. A potentially relevant explanatory factor in this regard is trust, which is believed to increase compliance with governmental rules and regulations and sustain confidence in information provided by authorities [24–28]. Moreover, trust in health care is important to sustain the legitimacy of the health care system in general, as well as to maintain adherence to regulations and motivate patients to accept health care interventions [29]. Moreover, the results from the aforementioned pilot study [22] indicate trust in health care as a possible motivational driver worth more extensive inquiry.

Several studies bring up trust as potentially significant for antibiotic use [30–32], and some assess this assumption empirically. Touboul-Lundgren and colleagues include trust in medical doctors as an integrated part of their indicator of culture [32]. Other studies suggest that practitioners are concerned about losing patients' trust if they do not prescribe antibiotics [33, 34]. In terms of explanations for between-country variation in antibiotic consumption, Blommaert and colleagues [34] found social trust–trust in other people–to be linked to country-levels of antibiotic consumption. A few studies pay attention to the individual-level relationship between trust and attitudes towards antibiotics and antibiotic use. In survey data research for Sweden, both social trust and trust in health care have been found to be positively linked to willingness to limit antibiotic use [35, 36]. Carlson and colleagues [37] found that trust in physicians was associated with acceptance of doctors' decision not to prescribe antibiotics. In view of these findings and given that trust in institutions is linked to confidence in information and decision acceptance, the potential role of trust in health care as constraining people's willingness to acquire antibiotics without a prescription warrants scrutiny.

We therefore conducted a survey of a sample of the Swedish population (representative in terms of age, gender and education), with previous non-prescription acquisition of antibiotics and willingness to engage in this practice in the future as primary outcomes. Non-prescription acquisition was defined as acquiring antibiotics in other ways than from a Swedish physician issuing a prescription. The outcome variables were analytically linked to each other and to background variables considered potentially interesting, including trust in health care. Motives and sources of non-prescription acquisition were also examined.

## 2. Materials and methods

### Pilot study

The aim and design of the present study comes out of the aforementioned pilot study [22]. This study indicated two particularly important results. First, that acquisition of antibiotics without prescription may undermine the effectiveness of Swedish AS strategies. Second, that motives relating to people's trust in health care may be a driver of their antibiotics acquisition behavior. The pilot study included a mix of preset response alternatives and open qualitative questions, and the responses to the latter grounded the design of the preset response alternatives in the survey of the present study. They also grounded our aim to test the specific hypothesis that trust in health care may explain antibiotics acquisition behavior. This since responses

to the open questions in the pilot study strongly indicated distrust in health care as a main motivation among the respondents reporting that they had or were prepared to acquire antibiotics without a prescription. While the pilot study was exploratory and therefore did not control for bias in collected data and employed a mix of quantitative and qualitative method, the present study was designed to assess the magnitude of antibiotics acquisition behaviors and attitudes related to prescription in the Swedish population, focusing on trust in health care as a main potential explanatory factor of these behaviors. Since one's personal view of one's own health and one's general health care seeking behavior are a priori confounder-candidates for any specific health service seeking attitude and behavior, and also weakly signaled among the motives described in the pilot, we added these as further explanatory factors to be tested alongside the trust hypothesis. Likewise, we added education level as a possible explanatory factor (besides a background variable used to stratify the sample together with age and gender, see below), since this factor is often considered a potential explanation of differences in health service seeking attitudes and behaviors in general.

## Data collection

Data were collected through an online survey via the Citizens Panel, administered by the SOM Institute, University of Gothenburg, Sweden. The Citizens panel consists of a pool of self-recruited respondents who answer questions about four times a year without remuneration. The respondents are invited to participate in each survey wave via email, and a maximum of three reminders are sent out during the field period. The respondents had to provide informed consent before they could take the survey. The number of respondents who did not consent and thus declined participation was 23 respondents (0,54 percent).

The survey was open between 15 September and 26 October 2020. The sample was pre-stratified to mirror the Swedish population in terms of gender, age and education. Alongside questions regarding antibiotics and antibiotic use, the survey also contained questions about respondents' health care contacts, self-rated health and trust in health care (see S1 Appendix).

## Outcome variables

The two outcome variables were 1) previous acquisition of antibiotics other than from a physician in Sweden having issued a prescription and 2) likelihood of future acquisition of antibiotics other than from a physician in Sweden issuing a prescription. The survey question used to capture the first outcome variable was: "In the last five years, have you obtained antibiotics without a prescription from a physician in Sweden?". The response alternatives were "Yes", "No" and "Don't remember". The second outcome variable survey question was: "If you or a close relative were to become ill at some point during the next five years, how likely is it that you would obtain antibiotics without a prescription from a physician in Sweden (regardless of the cause)?". The response alternatives were: "Very likely", "Fairly likely", "Fairly unlikely" and "Very unlikely".

## Explanatory variables

Trust in health care was measured by the question "How much trust do you have in Swedish health care?". Five response alternatives were given: "Very much trust", "Fairly much trust", "Neither much nor little trust", "Fairly little trust", and "Very little trust".

Self-rated health was measured using the following question: "How would you rate your general health status?". The response alternatives ranged on a scale from 0 (very poor health) to 10 (very good health).

The education variable was built upon in total nine survey response categories. "Low education" = completed or incomplete 9-year elementary school, "Medium–low" = gymnasium/similar < 3 years or ≥ 3, "Medium–high" = post gymnasium studies other than university education < 3 years or ≥ 3, and "High education" = university education < 3 years, ≥ 3 years or graduation from third-cycle university education.

## Ethics

The study processed potentially sensitive personal information about the respondents and therefore required legally mandatory ethics review, secure storage of raw data and standard collection of informed consent. All members of the Citizen Panel had given generic informed consent to be approached for studies, and those invited to participate in the present one were given specific information about the study before consenting (or declining) to participate. All results below are presented on a generic group level. The study received authorization by the Swedish Ethical Review Authority (no. 2020–03019) before initiation.

## Statistics

Two multiple logistic regression models were used to investigate the association between the explanatory variables and the two dependent variables: 1) previous non-prescription acquisition of antibiotics (Tables 2 and 3) likelihood of future non-prescription acquisition of antibiotics (Table 4). Tables 3 and 4 reports Odds ratios (ORs) and 95% confidence intervals (CIs). All data were analysed with the STATA 17 statistical software package.

## 3. Results

The survey yielded 4243 responses and the participation rate was 57%. We calculated the participation rate by the quotient of the number of respondents who completed more than 50 percent of all questions in the survey and the number of respondents in the initial sample. Among the respondents who completed the survey, the gender balance was very equal– 51 percent were women and 49 percent men. The share of respondents who were 70 years or older when completing the survey was 17 percent and the share of respondents with three years of university education or more was 27 percent. This is reasonably similar to the Swedish population over 18 years (2020), where the 70+ group was 13 percent of the population and the share of people with university education (three years or more) was 24 percent.

### Descriptive and bivariate

In total, 2,3% (n = 97) answered that they had obtained antibiotics in other ways than from a doctor in Sweden having issued a prescription over the last five years. 43% of these answered that they had received antibiotics via a prescription by a physician abroad, 28% had bought antibiotics abroad without a prescription, 15% had received antibiotics from a relative or friend, and 9% had bought it via an online pharmacy without prescription.

About one in four answered that the most common reason for non-prescription acquisition was that they knew what kind of antibiotics they needed (26%). About one in five wanted to stockpile antibiotics in case they were to be denied antibiotics by a physician (18%) and 14% because it is easier to obtain antibiotics without than with a prescription. Less than 10% stated that the reason for non-prescription acquisition was that a physician had denied them antibiotics (5%), or that they expected to be denied a prescription (8%). The largest group selected the open-ended response alternative which gave the respondents the possibility to write their

own words (53%). Almost all of these answers concerned the use of antibiotics while being abroad.

The survey also contained a question about whether respondents believed that they would obtain antibiotics in other ways than from a physician in Sweden issuing a prescription in the future should they or a relative become ill. Out of the four response alternatives provided, the two most common responses were "Very unlikely" (75.8%, n = 3 176) and "Rather unlikely" (19.9%, n = 833). About 2.5% of the respondents answered "Somewhat likely" (n = 103) and 1.9% "Very likely" (n = 80).

The likelihood of future non-prescription acquisition of antibiotics differed depending upon previous non-prescription acquisition. Among the respondents who had previously obtained antibiotics in other ways than from a physician in Sweden issuing a prescription, about 30 percent reported that they were likely to do so in the future (30 out of 97 respondents). This share is less than 4 percent among respondents who responded not having obtained antibiotics in such ways before (153 out of 4081).

Table 1 reports responses to the question of reasons for possible future non-prescription acquisition of antibiotics. The question was: "How important or unimportant are the following reasons why you would probably buy antibiotics without a prescription from a doctor in Sweden?". This question was only asked to respondents who in the previous question stated that it is "Somewhat likely" or "Very likely" that they will obtain antibiotics in such a way in the future (n = 183). The number of respondents who responded to the questions about reasons for future non-prescription acquisition of antibiotics varied between 161–167.

The table shows that the most important reason was that respondents expected physicians not to prescribe antibiotics. On the scale from 1 (Very unimportant) to 5 (Very important), the means score for this reason was 3.1 (median 3). The second most important reason was obtaining antibiotics via prescription from a physician abroad (mean 3.0, median 3). Reasons such as ease of acquisition (mean 2.6, median 2), availability abroad (mean 2.6, median 3) and stockpiling to keep antibiotics as reserve (mean 2.5, median 2) are in the middle in terms of importance. Less important reasons are avoiding getting in contact with a physician (mean 2.1, median 1) and ability to decide about antibiotics oneself (2.4, median 2).

**Table 1. Reasons for non-prescription acquisition of antibiotics in the future (mean, 95% confidence interval, median, and number of respondents).**

| | Mean (1–5) and Confidence interval (95%) | Median | n |
|---|---|---|---|
| Physicians don't want to prescribe antibiotics | 3.1 (2.9–3.4) | 3 | 167 |
| Avoid getting in contact with a physician | 2.1 (1.9–2.3) | 1 | 161 |
| I can decide for myself if I or a close relative/child need antibiotics | 2.4 (2.2–2.7) | 2 | 163 |
| I am abroad and can get antibiotics without a prescription | 2.6 (2.4–2.9) | 3 | 165 |
| I am abroad and can get antibiotics via prescription from a doctor there | 3.0 (2.8–3.2) | 3 | 166 |
| To keep antibiotics at home as a reserve in case I do not get it prescribed by physician | 2.5 (2.3–2.8) | 2 | 166 |
| Because it's easy | 2.6 (2.3–2.8) | 3 | 164 |

**Comment:** Question wording: "How important or unimportant are the following reasons why you would probably buy antibiotics without a prescription from a doctor in Sweden?" The scale in the survey was from 1 (Very important) to 5 (Very unimportant), but it is reversed in the presentation of data (1 = Very unimportant, 5 = Very important) **Source:** The Citizens Panel wave 39.

**Table 2. Reasons against engaging in non-prescription acquisition of antibiotics (mean, 95% confidence interval, median, and number of respondents).**

| | Mean (1–5) and Confidence interval (95%) | Median | n |
|---|---|---|---|
| To avoid getting worse health | 3.95 (3.91–4.00) | 5 | 3880 |
| In order not to contribute to antibiotic resistance | 4.64 (4.61–4.66) | 5 | 3910 |
| The quality of over-the-counter antibiotics feels unsafe | 4.48 (4.45–4.51) | 4 | 3911 |
| I trust that physicians can decide if I need antibiotics | 4.53 (4.50–4.56) | 4 | 3911 |
| There is a risk of incorrect dosing (taking too much or too little of the antibiotic) | 4.19 (4.15–4.23) | 4 | 3913 |
| Prescriptions are safer due to pharmacists' control | 4.33 (4.30–4.37) | 4 | 3917 |

**Comment:** Question wording: "How important or unimportant are the following reasons why you would probably not buy antibiotics without a prescription from a doctor in Sweden?" The scale in the survey was from 1 (Very important) to 5 (Very unimportant), but it is reversed in the presentation of data (1 = Very unimportant, 5 = Very important) **Source:** The Citizens Panel wave 39.

Respondents rather or very unlikely to obtain antibiotics in other ways than from a physician in Sweden issuing a prescription were asked about the importance of different reasons for not doing so. The results are presented in Table 2 and show that the most important reason was to not contribute to antibiotic resistance (mean 4.64, median 5). Other important reasons were trust in physicians' judgment about whether antibiotics are needed (mean 4.53, median 4), that over-the-counter antibiotics are unsafe (mean 4.48, median 4) and the safety of pharmacists' control (mean 4.33, median 4). Reasons less important were risk of incorrect dosing (mean 4.19, median 4) and health risks (mean 3.95, median 4).

## The role of trust in health care

Fig 1 demonstrates non-prescription acquisition of antibiotics according to respondents' level of trust in Swedish health care. It shows that trust is strongly negatively linked to both having

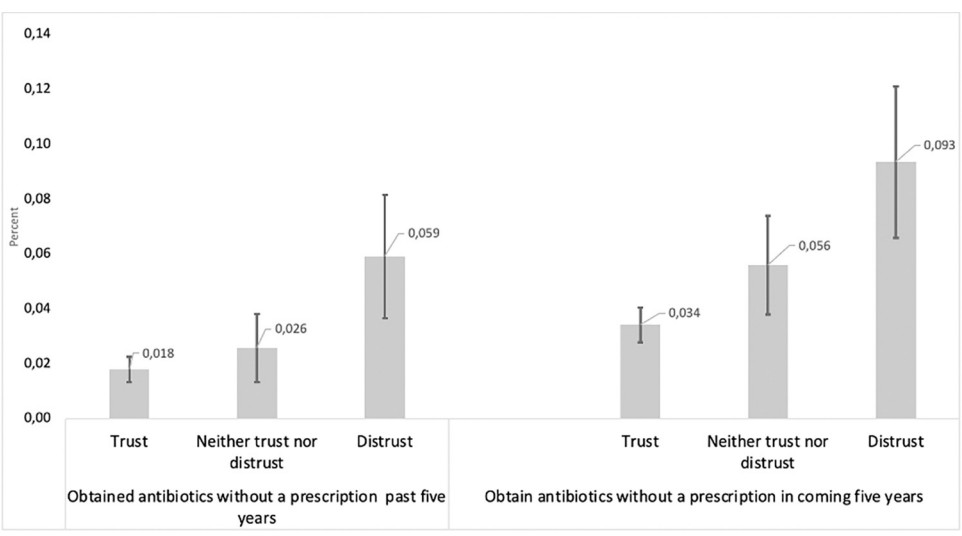

**Fig 1. Non-prescription acquisition of antibiotics according to trust in health care (percent 95% confidence interval). Comment**: Antibiotics acquisition question wordings: see Tables 1 and 2. Trust question wording: "How much trust do you have in Swedish health care?" The three trust categories are based on five response alternatives: "Trust" (fairly much and very much trust), "Neither trust nor distrust" (Neither much nor little trust) and "Distrust" (very little and fairly little trust). **Source:** The Citizens Panel wave 39.

engaged in this practice in the previous five years and intention to do so in the coming five years. Non-prescription acquisition is three-fold higher in the distrust group, as compared to the trust group.

## Regressions

Using logistic regression, Table 3 illustrates the association between the outcome variable previous five-year non-prescription acquisition of antibiotics and the explanatory variables sex, age, education, self-rated health, trust in health care, use of antibiotics last year and health care contacts last year.

The table shows no significant associations between previous non-prescription acquisition of antibiotics and the explanatory factors sex, education, age, self-rated health and usage of prescribed antibiotic. There is a weak but significant association between contact with health care and non-prescription antibiotic acquisition; the likelihood for this is higher in the group of people that had not been in contact with health care the past year. Furthermore, there is a significant and strong association between trust in health care and previous non-prescription acquisition of antibiotics. The likelihood of this is significantly higher in the four groups "Fairly much trust", "Neither much nor little trust", "Fairly little trust" and "Very little trust",

**Table 3. Previous non-prescription acquisition of antibiotics, according to demographic variables, self-rated health, trust in health care, antibiotic use and health care contacts.** Odds ratios (OR), 95 confidence intervals (95% CI) and p-values (n = 4 169).

|  | Odds Ratio (OR) | C.I. (95%) | P-value |
|---|---|---|---|
| Woman (Ref. Cat.) | 1.00 |  |  |
| Man | 1.21 | 0.80–1.85 | 0.367 |
| 16–29 years (Ref. Cat.) | 1.00 |  |  |
| 30–39 years | 0.91 | 0.39–2.10 | 0.825 |
| 40–49 years | 0.71 | 0.30–1.67 | 0.430 |
| 50–59 years | 1.24 | 0.57–2.72 | 0.592 |
| 60–69 years | 1.40 | 0.64–3.07 | 0.396 |
| 70 years or older | 0.63 | 0.24–1.61 | 0.332 |
| Low education (Ref. Cat.) | 1.00 |  |  |
| Medium low education | 0.79 | 0.26–2.35 | 0.666 |
| Medium high education | 1.22 | 0.41–3.67 | 0.719 |
| High education | 1.14 | 0.40–3.28 | 0.805 |
| Self-rated health (0–10) | 1.01 | 0.91–1.14 | 0.801 |
| Very much trust (Ref. cat.) | 1.00 |  |  |
| Fairly much trust | 2.35 | 1.14–4.83 | 0.020 |
| Neither much nor little trust | 2.78 | 1.21–6.42 | 0.016 |
| Fairly little trust | 6.46 | 2.84–14.72 | 0.000 |
| Very little trust | 9.19 | 3.10–27.27 | 0.000 |
| No antibiotics last year (Ref. Cat.) | 1.00 |  |  |
| Antibiotics 1 time last year | 0.91 | 0.43–1.92 | 0.804 |
| Antibiotics 2 time or more last year | 1.36 | 0.48–3.85 | 0.564 |
| Care contact last year (Ref. Cat.) | 1.00 |  |  |
| No care contact last year | 1.61 | 1.00–2.60 | 0.049 |

**Comment:** The antibiotic use question was: "Have you used antibiotics (e.g., penicillin) prescribed by a doctor in Sweden in the last 12 months?". Health care contact was captured with the question "When was your last contact with health care regarding yourself?" **Source:** The Citizens Panel wave 39.

**Table 4. Likelihood of future non-prescription acquisition of antibiotics, according to demographic variables, self-rated health, trust in health care, antibiotic use and health care contacts.** Odds ratios (OR), 95 confidence intervals (95% CI) and p-values (n = 4 195).

| | Odds Ratio (OR) | C.I. (95%) | P-value |
|---|---|---|---|
| Woman (Ref. Cat.) | 1.00 | | |
| Man | 2.19 | 1.58–3.04 | 0.000 |
| 16–29 years (Ref. Cat.) | | | |
| 30–39 years | 1.36 | 0.72–2.60 | 0.347 |
| 40–49 years | 1.33 | 0.71–2.51 | 0.372 |
| 50–59 years | 1.25 | 0.66–2.36 | 0.501 |
| 60–69 years | 1.06 | 0.55–2.04 | 0.862 |
| 70 years or older | 1.31 | 0.68–2.53 | 0.424 |
| Low education (Ref. Cat.) | 1.00 | | |
| Medium low education | 0.78 | 0.38–1.58 | 0.484 |
| Medium high education | 0.89 | 0.431.86 | 0.762 |
| High education | 0.75 | 0.37–1.51 | 0.425 |
| Self-rated health (0–10) | 1.05 | 0.97–1.15 | 0.242 |
| Very much trust (Ref. cat.) | 1.00 | | |
| Fairly much trust | 2.00 | 1.23–3.25 | 0.005 |
| Neither much nor little trust | 2.92 | 1.67–5.13 | 0.000 |
| Fairly little trust | 5.16 | 2.87–9.26 | 0.000 |
| Very little trust | 5.18 | 2.16–12.39 | 0.000 |
| No antibiotics last year (Ref. Cat.) | 1.00 | | |
| Antibiotics 1 time last year | 1.29 | 0.80–2.09 | 0.293 |
| Antibiotics 2 time or more last year | 1.49 | 0.70–3.16 | 0.302 |
| Care contact last year (Ref. Cat.) | 1.00 | | |
| No care contact last year | 1.14 | 0.79–1.65 | 0.489 |

**Comment:** The antibiotic use question was: "Have you used antibiotics (e.g., penicillin) prescribed by a doctor in Sweden in the last 12 months?". Health care contact was captured with the question "When was your last contact with health care regarding yourself?" **Source:** The Citizens Panel wave 39.

as compared to the "Very much trust" reference category. The odds of previous acquisition of antibiotics not prescribed by a physician in Sweden is 9 times higher in the group with very little trust in health care compared to the very much trust category.

Table 4 shows the association between the dependent variable likelihood of future non-prescription acquisition of antibiotics and the explanatory variables sex, age, education, self-rated health, trust in health care, antibiotic use and health care contacts. The table shows that men are more likely to have the intention to obtain antibiotics in such a way in the future. There are no significant differences in the likelihood depending on age, education, self-rated health, antibiotic use and health care contacts. However, similar to previous non-prescription antibiotics acquisition, there is a significant and strong association between trust in health care and the intention to engage in non-prescription antibiotic acquisition in the future. The likelihood for such acquisition is higher for each "step down" on the trust five-point scale. The likelihood is about five times as high in the "Very low trust" category, as compared to the "Very high trust" reference category.

## 4. Discussion

Our study indicates that few people in Sweden acquire antibiotics in other ways than via a physician in Sweden issuing a prescription. This share is 2,3%. The share of survey respondents

who say they are likely to obtain antibiotics in such ways in the future is also quite small–slightly over 4% say that this is somewhat or very likely.

Most respondents who had acquired antibiotics without such prescription had received a prescription from a physician abroad (43%). Thus, the share of the Swedish population that has obtained antibiotics without a physician in Sweden or elsewhere having issued a prescription is about 1,4%. Those who indicate that they have bought antibiotics without a prescription abroad was 28%, indicating a share of the Swedish population of less than 1%. These results are in line with a recent study from Norway, indicating that purchase of antibiotics during travel abroad is rare in the Norwegian general population [21]. The same low prevalence of purchase of antibiotics during travel abroad was found in a survey study in Denmark in 2003 [38].

The analysis shows that trust in health care is an important factor when it comes to non-prescription acquisition of antibiotics. This is in line with previous research indicating that trust in physicians is linked to higher willingness to accept doctors' decisions not to prescribe antibiotics [37] and higher willingness to postpone antibiotic use [35]. One possible interpretation of the results is that when people are being denied antibiotics from a physician, people with low trust look for alternative sources of antibiotics.

While the current study follows much of the previous literature in making use of a uni-dimensional indicator of trust in the health care system [39, 40], theoretical arguments speak in favor of institutional trust being a multi-dimensional concept. Institutions can be trusted if they are competent, have the motivation and the opportunity to fulfill their tasks in a good way [41]. Given the results of the current study, an interesting avenue for future research could be to measure the dimensionality of trust in the health care system and analyze what specific components of trust that may be important in curbing the demand for the use of non-prescription antibiotics. One such factor could be the degree of trust in the competence characterizing the health care system or health care personnel. The perceived competence is likely to influence the degree to which patients think doctors' decisions not to prescribe antibiotics is reasonable and medically motivated, and this trust dimension may thus lower the demand for antibiotics without a prescription. Another trust component is the perception that an actor or institution has good intentions. This dimension of trust could potentially trigger a reciprocal and prosocial feeling of collective responsibility for the problem of antibiotic resistance. This way, people might want to contribute to the fight against antibiotic resistance.

The strong link between trust and non-prescription antibiotic acquisition is interesting in the light of the variation in trust between countries world-wide. The level of trust in health care is rather high in Sweden in a country comparative perspective [42]. It is therefore likely that non-prescription acquisition of antibiotics is higher in countries where the level of trust is lower. Furthermore, the results indicate that implementing prescription control policies against overuse may be difficult in a low trust context, since many more people will look for antibiotics elsewhere, for example via online pharmacies. What complicates this further is that low trust societies often are characterized by corruption and low quality of government, features that may complicate the implementation of AS programs in general [43].

Regarding AS programs, our results suggest, first, that maintaining high levels of trust in health care institutions is a critical factor for the effectiveness of prescription control. Where general trust levels are known to be low, there are reasons for AS programs to include trust enhancing measures to accompany prescription control measures. Merely denying patients antibiotics is not enough; rather these people need to be made sufficiently motivated not to acquire antibiotics from other sources. Second, the wide variations of trust patterns worldwide is a reason to assess the level of non-prescription acquisition of antibiotics, and further test the hypothesis of the role of trust for such acquisition, in other countries that apply AS. High levels

of non-prescription acquisition indicate that AS may be ineffective, and that additional measures to enhance trust in health care and/or to block pathways to such acquisition should be contemplated.

## Limitations

Our focus on acquisition rather than use, albeit motivated (see Introduction), means that we have not measured self-medication with leftover antibiotics previously obtained with a prescription. Similarly, there remains a possibility that antibiotics obtained without prescription are not used in some cases. Nevertheless, the phenomenon still highlights a challenge for AS through prescription control. The survey questions' simplified formulations in terms of physician prescription ignore the fact that some especially qualified specialist nurses may in some instances prescribe antibiotics in Sweden. However, almost all prescriptions of antibiotics in Sweden are made by physicians. Moreover, the motivations listed by respondents who stated that they had obtained antibiotics without prescription did not indicate any instance of nurse prescription. Given the generally high level of awareness about antibiotic resistance in Swedish society [44, 45]—including knowledge about the link between antibiotics overuse and resistance—there may be some underreporting due to preferences to avoid social stigma also in an anonymous online survey.

The results reported in this paper come from an online survey pre-stratified to mirror the Swedish population in terms of age, gender and education, and the sample is reasonably similar to the Swedish population with regard to these factors. However, one limitation in the data is that there might be other variables that would improve the estimates of behaviors and attitudes towards antibiotics, which were not included in the pre-stratification process.

## 5. Conclusion

The prevalence of non-prescription acquisition of antibiotics in the Swedish population is low, and the share of respondents intending to engage in this practice in the future is small. However, both past and intended non-prescription acquisition are strongly associated with low trust in health care. This highlights the importance of future studies of non-prescription antibiotics acquisition in low trust contexts and the need to consider trust preserving and enhancing measures when designing and implementing AS programs.

## Supporting information

**S1 Appendix. Questionnaire translated into English.**
(PDF)

## Acknowledgments

We are grateful to the preparatory work and participation in the early steps of the present study of Shadan Haidar Abdulla.

## Author Contributions

**Conceptualization:** Christian Munthe, Erik Malmqvist, Björn Rönnerstrand.

**Data curation:** Björn Rönnerstrand.

**Formal analysis:** Björn Rönnerstrand.

**Funding acquisition:** Christian Munthe.

**Investigation:** Christian Munthe, Erik Malmqvist, Björn Rönnerstrand.

**Methodology:** Christian Munthe, Erik Malmqvist, Björn Rönnerstrand.

**Resources:** Björn Rönnerstrand.

**Software:** Björn Rönnerstrand.

**Supervision:** Christian Munthe.

**Validation:** Christian Munthe, Erik Malmqvist, Björn Rönnerstrand.

**Visualization:** Björn Rönnerstrand.

**Writing – original draft:** Christian Munthe, Erik Malmqvist, Björn Rönnerstrand.

**Writing – review & editing:** Christian Munthe, Erik Malmqvist, Björn Rönnerstrand.

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
