## [Decision Letter · Decision Letter 0]

25 Feb 2022

PONE-D-22-02565Non-prescription acquisition of antibiotics: prevalence, motives, pathways and explanatory factors in the Swedish populationPLOS ONE

Dear Dr. Munthe,

Thank you for submitting your manuscript to PLOS ONE. After careful consideration, we feel that it has merit but does not fully meet PLOS ONE’s publication criteria as it currently stands. The study can be very interesting, the introduction is well written but the results and conclusions must be substantially implemented.The reviewers gave key directions for implementing the manuscript. Therefore, we invite you to submit a revised version of the manuscript that addresses the points raised during the review process.

We look forward to receiving your revised manuscript.

Kind regards,

Francesca Baratta, PharmD, PhD

Academic Editor

PLOS ONE

Journal Requirements:

“This study was funded by the UGOT-Challenges Initiative of the University of Gothenburg.”

 “Christian Munthe, no award no., UGOT Challenges Initiative, URL: https://www.gu.se/en/research/ugot-challenges 

Additional Editor Comments:

Dear Author,

The study can be very interesting, the introduction is well written but the results and conclusions must be substantially implemented.

The reviewers gave key directions for implementing the manuscript.

Best regards

Reviewers' comments:

Reviewer's Responses to Questions

**Comments to the Author**

1. Is the manuscript technically sound, and do the data support the conclusions?

Reviewer #1: Yes

Reviewer #2: No

Reviewer #3: Partly

2. Has the statistical analysis been performed appropriately and rigorously? 

Reviewer #1: No

Reviewer #2: Yes

Reviewer #3: I Don't Know

3. Have the authors made all data underlying the findings in their manuscript fully available?

Reviewer #1: No

Reviewer #2: Yes

Reviewer #3: Yes

4. Is the manuscript presented in an intelligible fashion and written in standard English?

Reviewer #1: Yes

Reviewer #2: Yes

Reviewer #3: Yes

5. Review Comments to the Author

Reviewer #1: This manuscript addresses the non-prescription use of antimicrobials in Sweden from the perspective of the issue of trust or lack of it in the healthcare system.

Overall, a well conceptualized and written manuscript

Main concern is the use of mean as measure of central tendency with Likert-like responses such as Very much trust”, “Fairly much trust”, “Neither much nor little trust”, “Fairly little trust”, and “Very little trust. Using the median will be more appropriate in these cases. This will apply to Tables 1 and 2.

Table 3 and are too bulky. The authors should kindly format according to the journal's specifications.

Limitations: I am not sure if "limitations should come under "Results". The authors to kindly check the journal's template/ submission guidelines

Reviewer #2: I think this paper has some interesting perspectives measuring the extent of acquiring antibiotics (AB) without prescription in a Northern European country. Even so, one would think there is no problem to explore, I think the authors do a good job in the introduction arguing for why such a study is still needed.

I think, however, there are several major problems with the study. This include: 1) lack of conclusion of the relevance of the study, 2) lack of methodological rigor, and 3) mis-/over-interpretation of the concept ‘trust in health care’, why I don’t think the study is ready to be published. Perhaps it can be considered a pilot study, showing an overall association between low trust in health care (whatever that is) and acquisition of AB that can be further explored in more studies (if one can justify such a study); or go amore in a more methodological direction and discuss in detail how the survey might be transformed to be used in other health care settings. See the comments below.

1: I think the authors fail really to discuss if the extend of acquiring AB as they measure it, proves if there really is a problem, that justifies spending time on the subject. In the introduction, the authors state p.5 that:

‘If a significant portion of people acquire antibiotics without prescription, the effectiveness of prescription control as a tool for AS would be undermined. Moreover, surveying prescription data to track the effect of AS on antibiotics use would provide a false picture that underestimates antibiotics use and overestimates the effectiveness of AS prescription control.’

However, I only see the authors making an indirect conclusion of whether they found a ‘significant portion’ of use of AB without prescription. Further, in the conclusion they don’t provide explicitly guidance to how the developed survey-tool could be used by authorities/ researchers from other countries:

‘The prevalence of non-prescription acquisition of antibiotics in the Swedish population is low, and the share of respondents intending to engage in this practice in the future is small. However, both past and intended non-prescription acquisition are strongly associated with low trust in health care. This highlights the importance of future studies of non-prescription antibiotics acquisition in low trust contexts and the need to consider trust preserving and enhancing measures when designing and implementing AS programs.’

2: I think the paper lacks rigorous theoretical and methodological underpinning. The authors develop if, I understand correctly, a completely new survey, however, there is no description of relevance of hardly any of the specific content of the survey. This includes a lack of explanation of most of the included questions and pre-determined nominal answers. This is in particular a problem since some answers such as the one: ‘How important or unimportant are the following reasons why you would probably buy antibiotics without a prescription from a doctor in Sweden?, only contains answers, predetermined by the authors. Further, there is no explanation of any validation process of this new developed survey. And lastly, there is no precise explanation of the sampling procedure.

Further, the key explanatory variable is ‘trust in health care’ or ‘institutional trust’. The authors explain that institutional trust can be measured with regard to trust in: government, agencies, or public institutions, and in relation to: if they are competent, have the motivation and the opportunity to fulfill their tasks in a good way. However, even so the authors describe the variable being multidimensional, only one question is asked in they survey: ‘How much trust do you have in Swedish health care?’ How do the authors know how respondents have interpreted this question, i.e. what their answers actually cover of all the possible dimensions.

3: In addition, despite the multidimensional construct of ‘trust in health care’, there seems to be an understanding among the authors in the discussion that this question in the survey covers respondents’ trust in physicians, which I find is a huge possible misinterpretation.

Further, the authors suggest that the developed survey could be used as a tool in other countries to show how low trust in health care are associated with high non-prescription AB acquisition. Here, I completely miss a discussion and literature of all the possible factors that might influence a difference in perception in ‘trust in health care’ between different cultural health care contexts, and therefore how you cannot repeat just this sole question on trust in the survey.

Lastly, there is no description of how the end responders represent or do not represent the Swedish population (including a more precise description of the sampling procedure), which however, doesn’t prevent the authors from making extrapolations of the results.

Reviewer #3: Summary

In their manuscript, “Non-prescription acquisition of antibiotics: prevalence, motives, pathways and explanatory factors in the Swedish population,” Munthe and colleagues describe a cross-sectional survey that asked about obtaining non-prescription antibiotics, sociodemographic factors, attitudes, and trust in healthcare.

The authors found that 2.3% of respondents reported obtaining non-prescription antibiotics, 4.3% are likely to do so in the future, and that prior receipt and intent to obtain non-prescription antibiotics in the future were associated with low trust in health care. People obtained antibiotics via prescription from a physician abroad (43%), bough antibiotics abroad without a prescription (28%), received antibiotics from a relative or friend (15%), or bought it through an online pharmacy without a prescription (9%).

General Comments

The manuscript is interesting, especially given the very low overall history and intent of receiving non-prescription antibiotics. The authors usefully frame their results as an example of what is possible in a high-trust environment. I especially appreciated the sentence “Merely denying patients antibiotics is not enough; rather these people need to be made sufficiently motivated not to acquire antibiotics from other sources.”

The study and manuscript have several weaknesses.

First, hopefully something is not getting translated accurately, but the wording of the main question in English is unclear. When I read the phrase “acquired antibiotics without a prescription from a Swedish physician” I took it to mean that Swedes were getting antibiotics directly from Swedish physicians, but without a prescription. Perhaps Swedish physicians were giving out sample antibiotics or had some other mechanism to give out antibiotics without issuing a prescription. (In a totally unscientific test, I asked my spouse how they interpreted the question, and they interpreted it the same way.)

It was not until page 11, at the beginning of the Results that includes the ways in which Swedes obtained non-physician-issued antibiotics that I understood the authors meant “acquired antibiotics, but not from a Swedish physician having issued a prescription.” Another way to word this in the Abstract and early in the manuscript is “acquired antibiotics, other than from a Swedish physician issuing a prescription.” With that, the authors’ meaning should be clear.

Second, the authors repeatedly state that the sample is representative, but the respondents are non-representative for at least two reasons. First, to even be included in the survey, potentially participants had to give informed consent. Second, the sample was “pre-stratified to mirror the Swedish population,” but there was a 57% response rate. Those willing to give informed consent and respondents are likely to be systematically different from Swedes who did or would not give informed consent and non-respondents (e.g., in their interest in the topic, willingness to report non-socially acceptable or illegal behavior, and other factors). I would guess consenting respondents would be less likely to acquire non-prescription antibiotics than the average person. Do the authors have any data about how respondents were demographically similar or different from the Swedish population?

Third, I would want the authors to report on the overlap of respondents who said they had obtained non-prescription antibiotics in the past and intend to do so in the future. Are these the same people? If so, this problem could be very limited. If not, the disconnect between past behavior and future intended behavior could be interesting for the authors to discuss.

Fourth, the authors never state what kind of statistical analysis they are using. Also, it is not clear how the authors are evaluating each variable in a model. I would think they should be using a Type 3 test or a test-of-trend for some of the variables, but there appear to be p-values for all levels of all variables.

Fifth, the data presented in the tables is difficult to interpret. The authors do not describe how importance was asked in the Methods section. Also, the Ns are hard to follow. For Table 1, each variable has an N of about 160 to 170, but 183 respondents said they were somewhat or very likely to obtain antibiotics in the future. Are these the same subsets? The tabular results need to more clearly state who is responding to each question.

Specific Comments

Page 4, 1st paragraph: The Introduction seems like a repeat of information that should be in an Abstract rather than an introduction.

Page 4-9: The Background and Introduction are very long. I am accustomed to a Background section that is 1-2 pages and gets right to the point of the study.

Page 9, paragraph 3: The authors need to explain the parenthetical “AAPOR RR5” and why they chose that method of calculating the response rate. Also, the reporting of responses and response rate seems like it belongs in the Results section.

Page 12, Table 1: The authors should clarify in the table title whether this refers to past behavior or future intent (I believe it is future intent).

Page 15, Table 3: There are some formatting problems with some of the confidence intervals. Perhaps the hyphen is missing?

Page 17, paragraph 2: The authors are incorrect in saying the odds ratio represents something is “9 times higher.” It is actually “9 times the odds,” which is different.

Page 20, paragraph 1: The authors should include data to their statement that Swedes have a “high level of awareness about antibiotic resistance.”

Figure: There is no key for the y-axis. I presume this is %.

6. PLOS authors have the option to publish the peer review history of their article (what does this mean?). If published, this will include your full peer review and any attached files.

Reviewer #1: **Yes: **Prof. Joseph O. Fadare

Reviewer #2: No

Reviewer #3: No

---

## [Author Response · Author response to Decision Letter 0]

7 Jul 2022

See the Cover letter and the response to reviewers files for details.

---

## [Decision Letter · Decision Letter 1]

3 Aug 2022

Non-prescription acquisition of antibiotics: prevalence, motives, pathways and explanatory factors in the Swedish population

PONE-D-22-02565R1

Dear Dr. Munthe,

We’re pleased to inform you that your manuscript has been judged scientifically suitable for publication and will be formally accepted for publication once it meets all outstanding technical requirements.

Kind regards,

Francesca Baratta, PharmD, PhD

Academic Editor

PLOS ONE

Reviewer's Responses to Questions

**Comments to the Author**

1. If the authors have adequately addressed your comments raised in a previous round of review and you feel that this manuscript is now acceptable for publication, you may indicate that here to bypass the “Comments to the Author” section, enter your conflict of interest statement in the “Confidential to Editor” section, and submit your "Accept" recommendation.

Reviewer #1: All comments have been addressed

2. Is the manuscript technically sound, and do the data support the conclusions?

Reviewer #1: Yes

3. Has the statistical analysis been performed appropriately and rigorously? 

Reviewer #1: Yes

4. Have the authors made all data underlying the findings in their manuscript fully available?

Reviewer #1: (No Response)

5. Is the manuscript presented in an intelligible fashion and written in standard English?

Reviewer #1: Yes

6. Review Comments to the Author

Reviewer #1: (No Response)

7. PLOS authors have the option to publish the peer review history of their article (what does this mean?). If published, this will include your full peer review and any attached files.

Reviewer #1: **Yes: **Prof. Joseph O. Fadare

---

## [Editor Report · Acceptance letter]

29 Aug 2022

PONE-D-22-02565R1 

Non-prescription acquisition of antibiotics: prevalence, motives, pathways and explanatory factors in the Swedish population 

Dear Dr. Munthe:

I'm pleased to inform you that your manuscript has been deemed suitable for publication in PLOS ONE. Congratulations! Your manuscript is now with our production department. 

Kind regards, 

on behalf of

Dr. Francesca Baratta 

Academic Editor

PLOS ONE